# Consumer Acceptance Studies of Margarine to Guide Product Development in the Food Industry

**DOI:** 10.3390/foods13010116

**Published:** 2023-12-29

**Authors:** Helena Maria Andre Bolini, Alessandra Cazelatto Medeiros, Cecília Teresa Muniz Pereira, Francisco Carraro, Pedro Pio Campregher Augusto, Flavio Cardello, Rafael Sousa Lima

**Affiliations:** Department of Food Engineering and Technology, University of Campinas, Monteiro Lobato, 80, Campinas 13083-862, Brazil; acmls@unicamp.br (A.C.M.); ceciteresa@ifma.edu.br (C.T.M.P.); carraro@unicamp.br (F.C.); pedrupio@gmail.com (P.P.C.A.); cardello1349@gmail.com (F.C.); nutrirafael@gmail.com (R.S.L.)

**Keywords:** consumer acceptance, margarine, texture, projective map

## Abstract

Margarine exhibits significant variations in composition, allowing it to cater to diverse consumer segments. This study aimed to characterize the physical and sensory attributes of margarine samples available in the Brazilian market. Twelve commercial samples from six different brands, encompassing 30% to 80% of lipid contents, were subjected to instrumental texture analysis and affective assessment. A total of 112 consumers participated in acceptance tests and Check-All-That-Apply (CATA) evaluations, while another group of 62 subjects performed Projective Mapping. Samples with lipid percentages exceeding 70% achieved the highest average acceptance scores for taste and overall impression. The brand with the lowest lipid content (30%) exhibited a stronger association with negative attributes, including rancid flavor and aroma, bitterness, and metallic flavor, resulting in lower average scores for aroma, flavor, texture, and overall impression. However, these scores were not statistically different from samples with 50% and 60% lipid content. Reducing lipid levels in fat-based products such as margarine poses a challenge to food manufacturers, as consumers generally perceive higher lipid percentages as indicative of superior flavor quality.

## 1. Introduction

Margarine is a water-in-oil emulsion consisting of an aqueous phase dispersed as fine droplets in liquid oil, usually derived from vegetable crops such as soybean, sunflower, corn, and palm, stabilized within a network of solid fat crystals [1]. During its production, the oil and aqueous phases are independently prepared and carefully mixed employing dosing systems to form an emulsion, WHICH is subsequently pasteurized and cooled to partially crystallize the fat and stabilize the system. In addition, other ingredients and additives may be added, i.e., salt, colorants, flavorings, and emulsifiers [2]. Initially, margarine was developed as a more affordable and versatile alternative to cow milk’s butter, as it resembles the latter in terms of texture, flavor, mouthfeel, and application. It is a globally appreciated product that is commonly used as a butter substitute in cooking, as an ingredient in baking, and as a table spread itself. Preferences for margarine over butter can differ due to cultural factors, personal tastes, culinary habits, and individual dietary requirements, as some people still prefer traditional butter [3].

The composition of margarine can vary substantially, leading to products with specific characteristics like low-fat, vitamin-enriched (principally vitamin A and D), or lactose-free options. In addition to appearance, flavor, and aroma, the consistency, mouthfeel, and texture of margarine are intrinsic properties that significantly affect their consumer acceptance [4]. Consistency is often a measure of smoothness, evenness, and plasticity, whereas texture refers mostly to structure, which can range from smooth to mealy, granular, crumbly, sandy, coarse, or lumpy. Margarine consistency and texture depend heavily on composition, fat content, the fatty-acid profile of the oil phase, and processing techniques [4,5]. These properties are key to consumer acceptance as they significantly impact mouthfeel [5]. 

Currently, there is a growing consumer concern about improving their diets for better health and well-being. As a result, a growing demand for well-balanced diets and functional foods that provide specific health benefits has been observed. Healthier versions of common foods include lower levels of sodium, sugar, and saturated fat, along with significantly reduced calorie density compared to their regular counterparts [6,7]. Consumer interest in low-fat margarine (containing less than 80% fat) has surged due to the detrimental impact of high saturated and trans-fat intake on cardiovascular health [8]. According to the latest Dietary Guideline for Americans [9], partially hydrogenated oils (PHOs), once a common ingredient of margarine and the major source of artificial trans-fat in foods, are no longer Generally Recognized as Safe (GRAS) by the Food and Drug Administration (FDA) as of 2018. In a previous publication, the US Department of Agriculture [10] advised American consumers to limit their daily consumption of fat spreads, especially hard or stick varieties, due to their high trans-fatty-acid content. Nonetheless, Weber et al. [11] found that most margarine brands in the US marketplace have already been reformulated to comply with the current daily trans-fat intake recommendation post-FDA ban on PHOs.

Manufacturers are constantly changing food products to meet the requirements of regulatory organizations and follow consumer trends. However, reformulations aiming to reduce the fat content in margarines may significantly affect their physical and structural properties, leading to undesirable outcomes for their sensory profiles and consumer acceptance. For instance, margarines with higher lipid contents are often associated with increased values of instrumental firmness and consumer acceptance scores for flavor and overall impression, in comparison to those with lower lipid contents. Additionally, margarines with lower lipid contents generally tend to exhibit negative sensory attributes more often, such as vegetable oil aroma, oily flavor, and rancid flavor, leading to lower consumer acceptance scores [4,12]. 

Sensory science comprises a set of systematic techniques aimed at accurately and objectively measuring human responses to consumer goods through their senses. As such, determining the sensory properties of the food itself provides key information for product development [13]. With the advancement of sensory analysis from the mid-twentieth century onwards, several sensory profiling methods have been developed. Some of them focus on training subjects based on standard sensory lexicons, whereas others enable subjects to more freely describe differences between samples (14).

Lately, rapid descriptive methods, i.e., Projective Mapping and its variants, Napping [14], and Check-All-That-Apply (CATA) [15], have gained popularity in the realm of sensory science. Among the advantages of these analyses is the possibility of tracing the sensory profiles of a large number of samples in significantly less time, in comparison to classical descriptive techniques such as Quantitative Descriptive Analysis (QDA^®^) [16]. 

Projective Mapping enables untrained subjects to group samples based on their similarities and differences by placing them on a two-dimensional surface using a piece of paper [17,18]. Subjects select their own criteria and determine their relative importance. Projective Mapping has been applied to various food products, including apples and cheeses [19], chocolate milk desserts [15], granola bars [20], wines [21,22], and spirits [23]. CATA enables sensory profile assessment using consumers, without the need for repetitions. It has been widely used because it is quick and simple to apply, in addition to presenting results similar to classical descriptive methods such as QDA^®^. This technique consists of selecting terms from a list of previously defined descriptors that best describe the product under evaluation. Data are analyzed according to the frequency of selection of each term through correspondence analysis and Cochran’s Q Test [18,24].

Given the difficulties of reducing lipid content in margarine without compromising some of its characteristic properties, the development of low-fat alternatives conforming to consumers’ expectations and regulatory standards often becomes a challenge for the food industry [25]. In this context, sensory science techniques emerge as valuable tools to monitor consumers’ reactions to margarine with varying lipid contents. 

All analyses in this study provide important information about margarine and a set of informative and leading results. It is relevant to note that the main focus of the present research was to apply consumer studies to obtain results relative to this perspective. However, it is important to highlight that further investigations are required, particularly regarding the chemical composition and fatty acids of the analyzed products.

The present work aimed to determine the textural properties, sensory profiles, and hedonic ratings of commercial margarine with varying lipid contents from the Brazilian market through rapid sensory descriptive techniques (CATA and Napping), consumer acceptance tests, and instrumental analysis. 

## 2. Materials and Methods

### 2.1. Samples

Twelve commercially available table margarine samples from different manufacturers, yet sharing the same production date, were analyzed, including light, creamy, and regular versions. As shown in Table 1, samples had differing fat contents, representing the various characteristics of margarines available in supermarkets in Campinas, Brazil. Noticeably, margarines with high fat content are more commonly found in the Brazilian market, as five of the samples had 80% lipids or more. All the samples contained salt in their formulations.

### 2.2. Instrumental Texture

Samples (4 °C) were evaluated in triplicate for instrumental texture using a TA XT2 texture analyzer (Systems, Godalming, UK) equipped with a 0.5 cm diameter cylindrical probe. The texture profile analysis (TPA) method was used through two sequential penetrations with a speed of 2 mm/s, a distance of 15 mm, and a force of 5 g. The following physical properties were evaluated: hardness, adhesiveness, cohesiveness, and resilience. 

Hardness is defined as the force required to deform or penetrate a food matrix. Regarding semi-solid foods, it refers to the force exerted by the tongue against the palate. Adhesiveness is related to the force required to break the force of attraction between the food matrix and other surfaces that come into contact with them, such as teeth. Cohesiveness refers to the strength of the internal connections that keep the integrity of a food matrix after a deformation event, as the extent of deformation of the food before its rupture is evaluated by measuring the difference between the forces of the first and second penetrations. Finally, resilience evaluates the recovery of the food structure after deformation, being the ratio between the area before and after deformation [26,27]. 

### 2.3. Sensory Analysis

Sensory analyses took place at the Laboratory of Sensory Science and Consumer Research (LSCCR) at the University of Campinas (UNICAMP), Campinas, SP, Brazil. The research was approved by the ethics and research committee of UNICAMP under the number CAAE 50531121.9.0000.5404. Subjects were invited to participate in the margarine study through specific social media groups created for this purpose and posters on university bulletin boards. They evaluated samples in individual booths with white light in a controlled environment at 21 ± 2 °C, following ISO guidelines [28]. Before any evaluation, subjects were asked about their willingness to participate in the tests as volunteers, and those choosing to collaborate signed an informed consent form. 

#### 2.3.1. Sample Preparation and Presentation

Samples were stored under refrigeration conditions at 4 ± 2 °C until the sensory analyses. Immediately before evaluation, samples were retrieved from the cold storage and served to the subjects. Moreover, subjects were provided with filtered water and Bauducco^®^ cracker biscuits (Extrema, MG, Brazil) to be consumed as palate cleansers between each sample. Samples were served in plastic cups labeled with unique three-digit codes, with each sample portioning at 20 g. Sample presentation followed a sequential monadic order, arranged within a complete balanced block design as described by Meilgaard et al. [29]. For the evaluation of spreadability, subjects used disposable plastic spatulas to smear the margarine samples on slices of crustless white bread (Bimbo Brasil, Mogi das Cruzes, SP, Brazil), cut in dimensions of 5 cm × 3 cm × 1 cm.

The acceptance test and CATA were performed in two separate sessions on two consecutive days, as six samples were evaluated in the first session and the remaining six samples in the second session, amounting to twelve samples presented to subjects in a completely balanced block design with sequential monadic order [29]. As for Projective Mapping, all twelve samples were presented simultaneously to each subject in a single session, who were asked to compare and group them according to their similarities and differences [30], and the data were collected during the sensory test through Fizz Network Sensory Software v.2.47b (Biosystèmes, Couternon, France). 

#### 2.3.2. Acceptance Analysis

Acceptance analysis was performed with 112 subjects who declared themselves to be consumers of margarine, thus representative of the target audience. They were recruited based on their frequency of margarine consumption (at least three times per week). Subjects were asked to evaluate how they liked samples about their appearance, aroma, flavor, texture, and overall impression using a 9 cm unstructured scale, anchored at the left and right extremes by the terms “I disliked it very much” and “I liked it very much”, respectively [13,16]. The purchase intention of consumers was also evaluated using a 5-point scale, ranging from “I certainly would not buy” to “I certainly would buy” [29].

#### 2.3.3. Check-All-That-Apply (CATA)

Check-All-That-Apply analysis was applied to the same 112 subjects who participated in the acceptance analysis questionnaire [31,32,33]. They were asked to evaluate samples and choose the terms that best described them from a list of 20 predefined descriptors [34], which were: yellow color, light yellow color, aerated, bright, milk aroma, rancid aroma, vegetable oil aroma, butter flavor, milk flavor, rancid flavor, sweet taste, grass flavor, oil taste, metallic taste, salty taste, bitter taste, sour cream taste, light texture, homogeneous, and consistent. Descriptive terms were previously determined through the repertory grid method [35,36]. The sequence of terms in the list was balanced for each subject and kept the same for all samples [24].

#### 2.3.4. Projective Mapping (Napping)

Projective Mapping was carried out with another group of 62 subjects, specifically recruited for this analysis, which is considered enough in the literature [37]. They were aged between 28 and 54 years old, and 50% declared themselves females. Subjects were selected based on their frequency of margarine consumption (more than twice a week) and were previously familiar with the analysis procedure. Each session lasted approximately 40 min. 

All 12 samples were presented simultaneously to each subject. Data were collected through the Fizz software program. They were asked to position samples in a two-dimensional plane on the computer screen so that samples perceived as being similar were grouped closer to each other, whereas samples perceived to be different were grouped farther from each other [38]. Subjects were also asked to list the characteristics of each group of samples to describe the sensory differences and similarities between the margarines [39].

As a result, each subject provided two coordinate vectors of 1 × 1 dimension each (one for the X axis, one for the Y axis), where they denoted the number of stimuli to be positioned in the rectangle. Thus, the final analyzed data set (denoted by X) was obtained by merging the N pairs of coordinate vectors, where N represented the number of subjects. In other words, X can be seen as a set of data structured into N groups of two variables each. Typically, the statistical analysis of data set X considers the “natural” partition of the variables.

### 2.4. Statistical Analyses 

Statistical analysis of data from the acceptance test and instrumental texture was performed by ANOVA and Tukey’s Honestly Significant Difference (HSD) Test of Means (*p* ≤ 0.05) using SAS software version 9.4 (Statistical Analysis System, Raleigh, NC, USA). Data from purchase intention were plotted in frequency histograms for each sample [29]. Data from CATA and Projective Mapping were analyzed using XLSTAT software version 2023 (Addinsoft, Paris, IF, France). Cochran’s Q test (*p* ≤ 0.05) and correspondence analysis (CA) were also performed for data from CATA and correlated to acceptance test data.

Furthermore, data from CATA and acceptance tests were correlated by employing Partial Least Squares (PLS) regression analysis [24]. The overall impression was considered the dependent variable (Y-matrix), while CATA parameters were the independent variables (X-matrix) [40]. Data from Napping were analyzed using Multiple Factor Analysis (MFA) and Hierarchical Cluster Analysis (HCA) [38,41]. 

## 3. Results and Discussion

### 3.1. Instrumental Texture Analysis

According to the results of instrumental texture analysis displayed in Table 2, samples with low fat contents were overall softer than those with high fat contents, although the latter did not differ significantly for hardness from samples D70 and C80, both having high lipid levels (70 and 80%, respectively). The hardest samples were A60, A80, and B80. Margarine quality is generally defined by its texture, consistency, hardness, and plasticity, with hardness being the most significant textural property perceived by consumers [42]. 

Adhesiveness is defined as the ease of removing margarine from the mouth or any surface [43]. Low-fat samples were significantly less adhesive (*p* ≤ 0.05) than those with high lipid content, except for B35, which did not statistically differ from the formers. Among high-fat samples, only A80 and F80 significantly differed (*p* ≤ 0.05) in cohesiveness from low-fat samples, those being the least cohesive of all samples. Regarding resilience, samples were segmented into two groups. High-fat samples showed lower values (A80, B80, C80, D82, A60, D70, and E50), whereas samples with the lowest lipid levels displayed significantly higher values (A30, B35, C38, and D55). Likewise, Silva et al. [44] verified similar behavior in relation to hardness and adhesiveness when analyzing margarines with different lipid contents. On the other hand, Ergönül [42] did not find statistical differences in the textural properties of hardness, adhesiveness, or cohesiveness when analyzing margarines sold in the Turkish market.

### 3.2. Acceptance Test

Acceptance ratings for appearance, aroma, flavor, texture, and overall impression are presented in Table 3. Means marked with the same letters in the same column do not significantly differ from each other, according to Tukey’s test. For attributes of appearance, aroma, and texture, no significant difference (*p* ≤ 0.05) was observed between samples regarding their fat contents. When considering means of overall impression, which is often indicative of subjects’ general hedonic response, high-fat samples (F80, D82, B80, A80, and D70) were preferred over those with lower lipid levels, except for sample C38, which did not significantly differ (*p* ≤ 0.05) from the latter. 

According to Andersen et al. [45], flavor presents the strongest correlation with the overall impression among all attributes of a product, which was also observed in this study, as samples that were preferred also scored the highest ratings for flavor. Exceptionally, sample C38 was significantly different from the sample with the highest mean for flavor (F80), despite being among the most preferred. Moreover, it is possible to confirm consumers’ preference for samples C80 and F80 by analyzing the histograms of purchase intention (Figure 1), while samples A30 and E50 scored the lowest purchase intention ratings by subjects. 

Acceptance test results are in line with the observations of Pădure [5], since there was a significant positive linear correlation between fat content and overall impression ratings, with 75.6% of the increase in acceptance being explained by the increase in the percentage of lipids in margarines (*p* < 0.05). Acceptance means relative to flavor exhibited a similar trend, as 74.5% of the increase in this variable was explained by the increase in the fat content of the margarine samples studied (*p* < 0.05).

### 3.3. Check-All-That-Apply (CATA)

Table 4 shows the frequency with which CATA terms were used to describe samples, as well as Cochran’s Q test p-values. According to Cochran’s Q test, 18 of the 20 attributes presented were considered significant (*p* < 0.05) to differentiate samples, which illustrates how consumers perceive variations in sensory characteristics based on the percentages of lipids in margarine from different manufacturers. Only the attributes “aerated” and “sweet taste” did not present any differences; that is to say, they did not contribute to the discrimination between samples. Furthermore, samples with lower lipid percentages exhibited a reduced frequency of mentions for “butter flavor”, while samples with higher lipid percentages exhibited a higher frequency of mentions for both “milk flavor” and “butter flavor”. The information obtained from this study could be valuable for the development of innovative products within the margarine and vegetable fat spreads industry, as well as in other segments that incorporate these products into their formulations.

The Principal Coordinates Analysis of CATA in Figure 2 shows the grouping of samples according to their similarities and differences, as well as the descriptors that characterize them the most by proximity, with an 81.87% explanation. The sample with the highest lipid content (D82) was described by the terms “texture”, “aeration”, “shine”, and “softness”. The other high-fat samples (A80, C80, F80, and B80) were correlated with the descriptors “consistent”, “butter flavor”, “milk aroma”, “milk cream flavor”, “milk flavor”, and “homogeneous”. Samples with intermediate lipid contents were characterized by “yellow color” and “salty taste”, whereas those with lower lipid levels were more related to the terms “rancid aroma”, “rancid flavor”, “metal flavor”, and “bitter taste”. Similar findings were observed by Foguel et al. [46] in commercial samples of cream cheese. The authors point out that the sample with the highest fat content was related to the descriptors “milky”, “mild flavor”, and “firm”, and the sample with the lowest fat content was more related to “sandy”, “matte”, “smooth”, “spreadable”, and “white”.

The Partial Least Squares (PLS) regression analysis shown in Figure 3 correlates the data from CATA with the hedonic data obtained in the acceptance test. Figure 3a displays which descriptors are directly related to the greater acceptance of the product, located close to the overall impression, those being “consistent”, “homogeneous”, “soft”, “bright”, “milk aroma”, “milk flavor”, “milk cream flavor”, and “butter flavor”, which are the same terms that best describe high-fat samples. Figure 3b, which illustrates the descriptors impacting positively (blue) and negatively (red) overall impression ratings, confirmed that “vegetable oil aroma”, “oily flavor”, and “rancid flavor” had a negative impact upon acceptance, as in the case of sample A30, the least preferred and best described by these descriptors.

These results are key for comprehending products’ overall acceptance. Notably, samples with the highest number of citations for specific characteristics had the highest lipid content, while those with the lowest citations had the lowest lipid content. These findings support the results in Figure 3, where it becomes evident that descriptors like “butter flavor” and “milk flavor” significantly influence preference for margarines.

Overall, fat plays a key role in flavor release by conveying taste- and odor-active compounds to chemosensory receptors as it melts during oral processing [46]. This may explain why samples with a lower percentage of lipids were the least preferred by subjects.

### 3.4. Projective Mapping (Napping)

Projective Mapping was used as a complementary tool in sensory profiling, considering the vast number of samples. Employing this method, samples were characterized by descriptors defined by the subjects themselves during evaluation. The first factorial plane issued by the Multiple Factor Analysis (MFA) and shown in Figure 4 indicates the similarities and differences between samples as perceived by subjects. Consumers generally differ in their sensory likes and dislikes, and this inter-individual variation can be reduced by organizing the population by the sensory preference segmentation method [47].

The dendrogram in Figure 5 shows the results of the Hierarchical Cluster Analysis (HCA), in which the samples were segmented into three distinct groups according to their similarities and differences. Cluster 1 is formed by seven samples (A80, B80, D82, B80, F80, D70, and D55) belonging to the group with the highest lipid content; cluster 2 is formed by C80 and B35; and cluster 3 is formed by A30, C38, and E50. Interestingly, cluster 2 samples have contrasting lipid contents, as there is a margarine with 80% fat and another with 35% fat content. This result originates from consumer data and can be attributed to the variability of products on the market, as each manufacturer can develop formulations with different sensory properties based on the type of raw material and/or added additives, such as preservatives or texturizers.

HCA and Napping results corroborated the sensory profiles obtained in CATA analysis, as Figure 4 shows that samples A80, B80, and F80 presented similarities in “mild aroma” and “margarine flavor”, whereas sample D82 was best described by the texture descriptor of “creamy”. Sample C80 was described by “milk flavor” and B35 was characterized by “unsalted”, whereas sample A30 was characterized by “watery flavor”, “residual”, and “artificial flavor”.

The influence of fat content on the sensory characteristics of margarines was clear. Samples (red bullets) are located near the descriptor (vector extremities) that best characterizes them, according to consumers. Samples with higher fat content were characterized mainly by “margarine flavor”, “creaminess”, “mild aroma” while those with lower fat contents were characterized by “watery flavor”, artificial flavor” and “rancid flavor”.

The findings of the present study are interesting and could be relevant to the area of food technology, especially relative to sensory science and consumer research. However, it is also important to consider new strategic approaches to the production of healthier margarines, such as the application of oleogel technology to totally or partially replace fat, which could be considered and discussed as a relevant perspective to advancements in the field.

CATA and Napping yielded different results, yet they were complementary to each other. However, it is important to emphasize that the results of these two methods rely on sensory lexicons sourced from differing databases. In CATA, participants receive the same predefined list of words, whereas in Napping, the set of analyzed words is formed spontaneously and individually by subjects based on their own perceptions and sensory references.

## 4. Conclusions

Results suggest that high-fat margarines tend to be harder and more adhesive, as well as more preferred, than their counterparts with lower fat contents, as shown by instrumental texture analysis and acceptance tests. However, the texture attribute seemed to be less critical than the flavor attribute to explain overall impression ratings. Furthermore, the sample containing only 30% of lipids, which was mostly associated with negative flavor descriptors, i.e., vegetable oil aroma, oily flavor, and rancid flavor, was also the least preferred among all samples, scoring the lowest averages for flavor, texture, and overall impression. These findings highlight the key role of fat in flavor release, as lipids act as carriers for tastants and odorants during the oral processing of fat-based products, especially edible water-in-oil emulsions. Several studies have dealt with the effects of lipid composition and concentration on the release rate of odor-active compounds that characterize flavor in various food systems [48,49,50,51,52].

This study also shows that it is essential to conduct focused sensory research aimed at obtaining margarines with reduced lipid levels, while still meeting consumers’ acceptance criteria. Additionally, conducting instrumental studies to evaluate the physicochemical characteristics of margarines in conjunction with sensory studies is recommended. This integrated approach can offer comprehensive insights to guide formulation adjustments and process enhancement, facilitating the development of nutritionally improved margarines while considering the potential effects of lipid reduction on consumer preferences.

## Figures and Tables

**Figure 1 foods-13-00116-f001:**
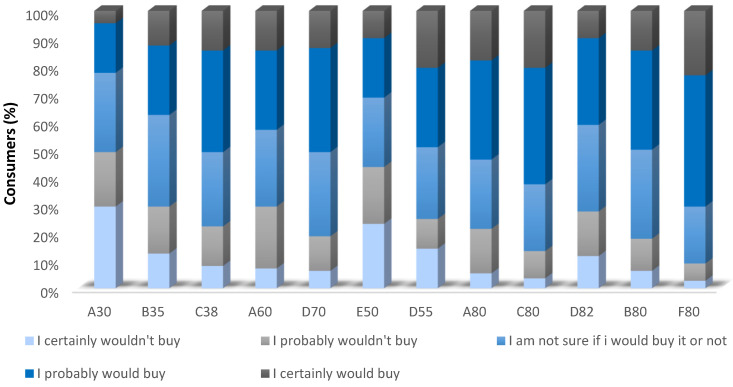
Histogram of purchase intention for margarine samples.

**Figure 2 foods-13-00116-f002:**
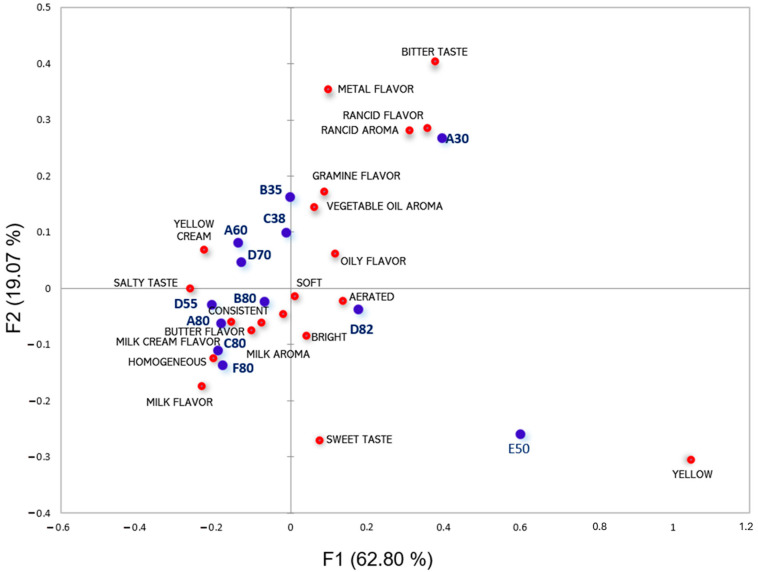
Principal Coordinates Analysis of CATA. Red bullets are the CATA descriptors, and blue bullets are the samples.

**Figure 3 foods-13-00116-f003:**
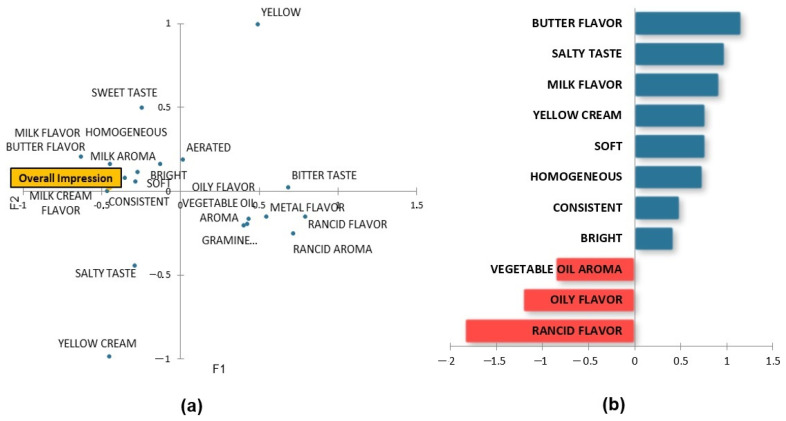
(**a**) Principal Coordinates Analysis. (**b**) Preference drivers of margarine.

**Figure 4 foods-13-00116-f004:**
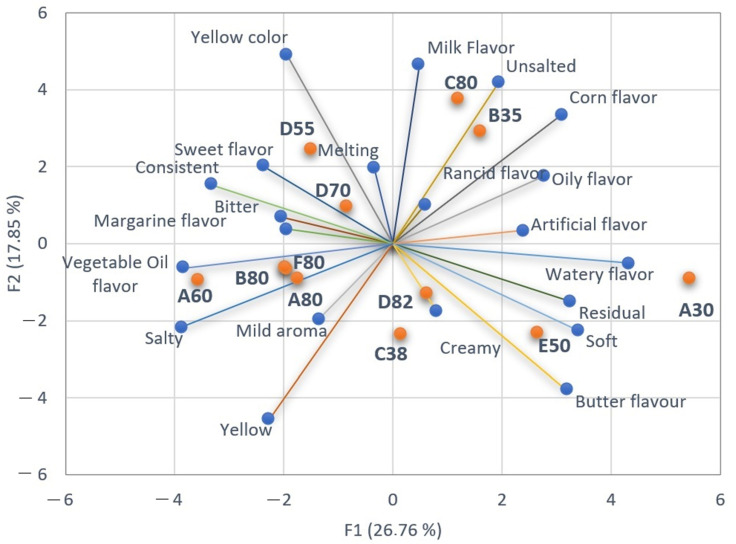
Multiple Factorial Analysis of Projective Mapping. Orange bullets are the samples, and blue bullets in the vector extremities are the descriptors of the samples.

**Figure 5 foods-13-00116-f005:**
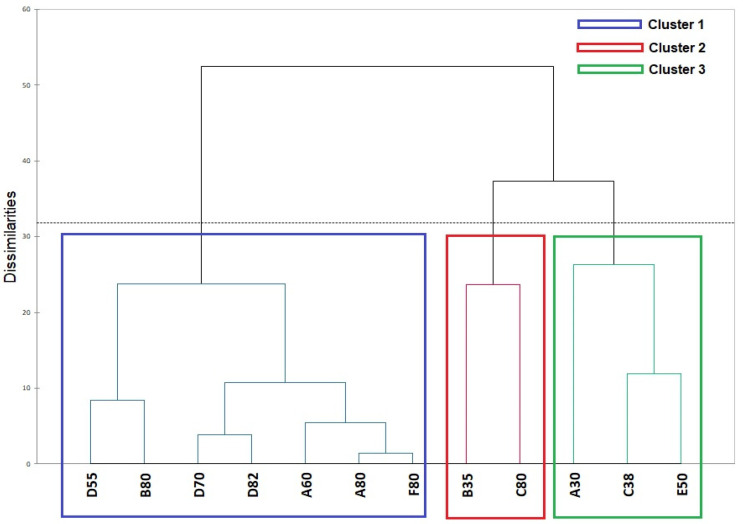
Dendrogram obtained through Hierarchical Cluster Analysis of Projective Mapping.

**Table 1 foods-13-00116-t001:** Samples coding and their respective nutritional information retrieved from labeling for a 10 g portion size.

Samples	Lipids (%)	Total Fat (g)	Saturated Fat (g)
A30	30	3.0	0.7
B35	35	3.5	1.2
C38	38	3.8	1.1
A60	60	6.0	1.8
D70	70	7.0	2.0
E50	50	5.0	1.4
D55	55	5.5	1.0
A80	80	8.0	1.8
C80	80	8.0	2.0
D82	82	8.2	2.3
B80	80	8.0	2.0
F80	80	8.0	2.0

**Table 2 foods-13-00116-t002:** Means of the parameters from the TPA curves for the different margarine samples.

Samples	Hardness (N)	Adhesiveness (Ns)	Cohesiveness	Resilience (Ns)
A30	168.008 ^a,b,^*	−688.280 ^c,d^	0.670 ^c,d^	0.023 ^e^
B35	209.665 ^a,b,c^	−976.496 ^b,c^	0.693 ^d^	0.016 ^c,d^
C38	120.477 ^a^	−539.837 ^d^	0.645 ^c,d^	0.015 ^b,c,d^
A60	523.980 ^g^	−2182.356 ^a^	0.601 ^b,c,d^	0.008 ^a^
D70	255.746 ^b,c,d^	−1230.185 ^b^	0.548 ^b,c,d^	0.010 ^a,b^
E50	304.365 ^d,e^	−1227.701 ^b^	0.618 ^b,c,d^	0.012 ^a,b,c^
D55	263.479 ^c,d^	−1159.119 ^b^	0.714 ^d^	0.020 ^d,e^
A80	735.396 ^h^	−1233.076 ^b^	0.270 ^a^	0.008 ^a^
B80	432.320 ^f^	−1971.677 ^a^	0.629 ^c,d^	0.007 ^a^
C80	259.202 ^b,c,d^	−1124.580 ^b^	0.538 ^b,c,d^	0.008 ^a^
D82	319.208 ^d,e^	−1128.141 ^b^	0.495 ^b,c^	0.007 ^a^
F80	370.914 ^e,f^	−1097.906 ^b^	0.432 ^a,b^	0.007 ^a^

* Means with the same letters in the same column do not differ statistically by Tukey’s test (*p* ≤ 0.05).

**Table 3 foods-13-00116-t003:** Means of acceptance in relation to appearance, aroma, flavor, texture, and overall impression.

Samples	Appearance	Aroma	Flavor	Texture	Overall Impression
A30	6.69 ^a,b,^*	5.35 ^b^	3.84 ^f^	5.73 ^c^	4.49 ^e^
B35	6.93 ^a,b^	5.98 ^a,b^	5.05 ^d,e^	6.41 ^a,b,c^	5.62 ^c,d^
C38	7.09 ^a,b^	6.41 ^a^	5.68 ^b,c,d^	6.89 ^a^	6.04 ^a,b,c^
A60	7.21 ^a^	6.54 ^a^	4.53 ^e,f^	6.58 ^a,b^	5.94 ^b,c,d^
D70	6.99 ^a,b^	6.27 ^a^	5.82 ^a,b,c,d^	6.12 ^b,c^	6.02 ^a,b,c^
E50	6.61 ^b^	6.22 ^a^	4.53 ^e,f^	6.57 ^a,b^	5.20 ^d,e^
D55	6.73 ^a,b^	6.16 ^a^	5.46 ^c,d^	6.12 ^b,c^	5.97 ^b,c^
A80	7.00 ^a,b^	6.70 ^a^	6.01 ^a,b,c^	6.89 ^a^	6.32 ^a,b,c^
B80	6.91 ^a,b^	6.28 ^a^	6.44 ^a,b^	6.83 ^a^	6.49 ^a,b^
C80	6.71 ^a,b^	6.21 ^a^	5.56 ^b,c,d^	6.37 ^a,b,c^	5.86 ^b,c,d^
D82	7.07 ^a,b^	6.24 ^a^	6.01 ^a,b,c^	6.70 ^a,b^	6.31 ^a,b,c^
F80	7.19 ^a,b^	6.16 ^a^	6.64 ^a^	6.91 ^a^	6.76 ^a^
MSD **	0.59	0.78	0.90	0.67	0.76

* Means with the same letters in the same column do not differ statistically by Tukey’s test (*p* ≤ 0.05). ** Minimum significant difference.

**Table 4 foods-13-00116-t004:** Frequency in which CATA terms were used to describe samples by subjects and Cochran’s Q sample comparison test.

	SAMPLES												*p*-Value (*p* < 0.05)
	A30	A60	A80	B35	B80	C38	C80	D55	D70	D82	E50	F80	
Yellow	43	6	8	14	20	16	8	2	4	39	83	12	<0.0001
Yellow cream	54	94	84	88	84	85	95	92	96	63	26	86	<0.0001
Aerated	11	10	9	9	5	14	13	9	6	5	14	7	0.081
Bright	56	58	58	47	60	53	67	48	58	67	67	74	<0.0001
Butter flavor	37	53	67	43	53	51	54	55	55	56	42	71	<0.0001
Milk aroma	15	23	32	17	26	15	23	26	21	20	14	22	0.010
Rancid aroma	32	10	11	11	7	11	7	16	15	16	14	7	<0.0001
Vegetable oil aroma	41	29	19	38	29	34	29	32	25	32	24	19	0.001
Milk flavor	14	22	44	14	22	17	36	41	30	20	19	34	<0.0001
Rancid flavor	52	29	23	23	15	27	7	10	22	28	28	13	<0.0001
Sweet taste	7	6	9	8	10	6	16	15	9	12	17	12	0.050
Gramine flavor	11	5	10	15	9	13	11	13	6	7	9	2	0.035
Oily flavor	79	57	50	62	58	60	51	57	62	65	66	58	0.002
Metal flavor	23	13	12	20	13	14	5	10	15	12	7	9	0.004
Salty taste	30	43	58	37	62	51	56	43	48	40	9	66	<0.0001
Bitter taste	19	1	4	12	9	9	6	4	9	7	8	2	<0.0001
Soft	65	66	61	63	69	78	81	62	69	58	70	70	0.014
Homogeneous	60	79	80	69	72	71	81	75	71	76	73	75	0.044
Consistent	25	43	50	46	44	40	41	46	39	42	38	46	0.013
Milk cream flavor	11	24	27	17	19	18	29	31	17	13	16	25	0.001

## Data Availability

Data is contained within the article.

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
