# Peer review of "Consumer Acceptance Studies of Margarine to Guide Product Development in the Food Industry"

_foods, 2023, doi:10.3390/foods13010116_

Round 1
Reviewer 1 Report
Comments and Suggestions for Authors
The title should be modified. The statement “.. Consumer Acceptance Studies to Guide Product Development in the Food Industry ..” is highly overstated. The quality of the manuscript is better now. The discussion of the results in the subsection 3.3 and 3.4 should be extended in the context of the influence of fat level on changes in the sensory characteristics of the products..
Author Response
Author's Reply to the Review Report: (Reviewer 1)
|
Point 1: The title should be modified. The statement “.. Consumer Acceptance Studies to Guide Product Development in the Food Industry ..” is highly overstated. Response 1: Dear Reviewer: We are thankful for all your comments and corrections, which contributed to the improvement of the manuscript. We accepted all of them, with gratitude. As suggested, the title was changed from Assessing the Influence of Fat Content Upon the Sensory and Textural Properties of Margarine to Guide the Development of Healthier Products To: Consumer Acceptance Studies to Guide Product Development in the Food Industry |
|
Point 2: The discussion of the results in the subsection 3.3 and 3.4 should be extended in the context of the influence of fat level on changes in the sensory characteristics of the products. Response 2: Thank you for your comments and suggestion, which contributed to the improvement of the manuscript. We accepted all of them. The iserted text is in red letter in the manuscript. |
Reviewer 2 Report
Comments and Suggestions for Authors
The Authors use different sensory and consumer testing methods to characterize and measure the perceptible properties of margarine samples. The introduction gives sufficient reasoning about the relevance of the subject. The chosen methods are suitable for the evaluation of the samples. The paper is well-written and properly constructed.
The sample set provides a sufficient range of the possible fat percentages of the margarines. In section 2.1. I miss the information regarding the type of margarine samples, whether these were table margarines, baking margarines or both types were involved. Later on (line 162) the text suggests that these samples were table margarines, because the spreadability was tested. The number of tested samples / sessions for the acceptance and CATA tests were 6, which meets the requirements of good sensory practice to avoid fatigue (ISO 6658 also refers on that). In case of Projective Mapping the simultaneous testing of all the 12 samples is generally accepted, in case of the rapid sensory methods, where the positioning of the samples and the distance among samples is a key criteria, we can work with a higher number of samples.
In the Abstract (line 18) we read about ‘sensory profiling’. In the methods section we do not read about sensory profile analysis or QDA. Please clarify the Abstract section in order to better understand, whether a sensory profiling was made or not.
At lines 203-204 there is a mention about capturing the position of the samples during Projective Mapping on the computer screen. Please provide additional information in the manuscript ‘Materials and methods’ section on the applied software during the sensory tests.
On Figure 5. the HCA analysis results are shown on Projective Mapping data. It is clear that on the left-hand side the majority of samples has higher fat content, while on the right-hand side we have lower fat content samples. However, there are some exceptions from this pattern. What is your opinion, why sample C80 is categorized to the group, where all the other samples have relatively low fat content?
Formal issues:
Line 548: at the end of the DOI there is a reference number, thus the link does not work this way. Please correct the DOI.
Line 570: The DOI of the last reference is doubled, please remove that part from the text.
Author Response
Dear Reviewer 2,
|
Point 4: On Figure 5. the HCA analysis results are shown on Projective Mapping data. It is clear that on the left-hand side the majority of samples has higher fat content, while on the right-hand side we have lower fat content samples. However, there are some exceptions from this pattern. What is your opinion, why sample C80 is categorized to the group, where all the other samples have relatively low fat content?.
Response 4: Thank you very much for the observation. I inserted a text about the formed clusters and the exception from the pattern.
|
|
Point 5: On Figure 5. the HCA analysis results are shown on Projective Mapping data. It is clear that on the left-hand side the majority of samples has higher fat content, while on the right-hand side we have lower fat content samples. However, there are some exceptions from this pattern. What is your opinion, why sample C80 is categorized to the group, where all the other samples have relatively low fat content?.
Response 5: Thank you very much for the observation. I inserted a text about the formed clusters and the exception from the pattern. Dear Reviewer, the Hieratchical Cluster Analysis originated three clusters of margarines related to the descriptors (or attributes) related to the margarines. I wrote in the text, and my opinion is that the of HCA is result is originated by consumer data, added to the variability of products on the market, as each margarine manufacturing industry can develop formulations with different flavors (based on the raw material and/or added additives, such as preservatives or texturizers). Thank you!
|
|
Point 6: Line 548: at the end of the DOI there is a reference number, thus the link does not work this way. Please correct the DOI.
Response 6: Thank you! Done.
|
|
Point 7: Line 570: The DOI of the last reference is doubled, please remove that part from the text.
Response 7: Thank you! Done.
|
|
|
Thank you!
Reviewer 3 Report
Comments and Suggestions for Authors
Dear Authors,
Your study, in which you analysed the variations in the composition among margarine samples available on the Brazilian market, offers insights into the relationship between lipid content and rheological characteristics, acceptability testing and projective mapping. The identification of negative attributes associated with lower lipid content, such as rancid flavour and aroma, bitterness and metallic flavor, deepens the discussion on consumer preferences. I commend the clarity with which you have presented your findings.. However, while your study takes a comprehensive approach, a more detailed look at the fats used in margarine production would have improved the understanding of the content of your results. Notably, there is no detailed analysis of the chemical composition of the products under consideration, such as proximate chemical analysis and fatty acid profiling. Existing literature emphasize the substantial influence of chemical composition (https://doi.org/10.3390/foods12051089) and, in particular, fatty acid profiles (https://doi.org/ 10.3390/foods11244054) on various sensory characteristics, ultimately impacting food acceptability. Based on these considerations, I give some suggestions: · In my opinion from the title of the article should be deleted “to guide the development of healthier products”. It seems advisable to align the title more closely with the specific contributions of your study. · In the text, it would be beneficial to specify that further investigations are needed particularly regarding the chemical composition and fatty acids of the analyzed products. This clarification would enhance the acceptability of your study · In discussions consider the use of new technologies in the margarin production (For instance, the use of Oleogel technology could be considered and discussed as a relevant aspect that aligns with advancements in the field.
I look forward to any additional information you may provide on the origin of the fat matter and its acidic composition in margarine making, as I believe this would further enrich the discussion and enhance the overall depth of your study.
Best regards

Author Response
Dear Reviewer, thank you for all considerations and suggestions. All of them are accepted and inserted in red text. Thank you
|
Dear Reviewer 3, Thank-you for your comments, suggestions. All of them contributed to the improvement of the manuscript. All text inserted is in red.
Point 1:. In my opinion from the title of the article should be deleted “to guide the development of healthier products”. It seems advisable to align the title more closely with the specific contributions of your study. · Response 1: Thank you. Correction done.
|
|
Point 2: In the text, it would be beneficial to specify that further investigations are needed particularly regarding the chemical composition and fatty acids of the analyzed products. This clarification would enhance the acceptability of your study
Response 2: Thank you. Correction done. A text in red letters is write before the last paragraph of the introduction.
|
|
Point 3: In discussions consider the use of new technologies in the margarin production (For instance, the use of Oleogel technology could be considered and discussed as a relevant aspect that aligns with advancements in the field.
Response 3: Thank you. The information was inserted in red letters before the last paragraph of Results and Discussion.
|